# Electrical Impedance Spectroscopy for Moisture and Oil Content Prediction in Oil Palm (*Elaeis guineensis* Jacq.) Fruitlets

**DOI:** 10.3390/plants11233373

**Published:** 2022-12-05

**Authors:** Nur Fizura Chin-Hashim, Alfadhl Yahya Khaled, Diyana Jamaludin, Samsuzana Abd Aziz

**Affiliations:** 1Department of Biological and Agricultural Engineering, Faculty of Engineering, Universiti Putra Malaysia, Serdang 43400, Malaysia; 2Department of Horticulture, College of Agricultural & Life Sciences, University of Wisconsin—Madison, Madison, WI 53705, USA; 3Smart Farming Technology Research Center, Universiti Putra Malaysia, Serdang 43400, Malaysia

**Keywords:** impedance spectroscopy, ECG, palm oil, oil content, moisture content

## Abstract

The global palm oil industry is targeting an increased oil extraction rate in oil palm milling to meet global demand. This can be achieved through the certification of mills and adherence to bunch grading as part of ensuring that only high-quality and ripe fresh fruit bunches are accepted and processed at all mills. However, the current grading process requires the analysis of oil palm bunches, which is laborious and tedious or prone to error due to human subjectivity. This paper introduces a non-destructive technique to predict the moisture and oil content in oil palm fruitlets using electrical impedance spectroscopy. In total, 90 samples of oil palm fruitlets at different stages of ripeness were acquired. Electrical impedance measurement of each fruitlet was done using electrocardiogram (ECG) electrodes connected to an LCR meter at frequencies of 1 kHz, 10 kHz, 20 kHz, and 100 kHz. The actual oil content in the fruitlets was determined using the Soxhlet extraction method, while the actual moisture content was determined using a standard oven-drying method. The variation of electrical impedance values at each frequency was analyzed. At 100 kHz, the correlation coefficients relating the electrical impedance to the moisture and oil content were around −0.84 and 0.80, respectively. Predictions of the moisture and oil content using linear regression of the impedance measurements at 100 kHz gave RMSE values of 5.85% and 5.71%, respectively. This information is useful for oil palm fruit grading and oil yield production estimation in the palm oil industry.

## 1. Introduction

The global palm oil industry is anticipated to reach a value of USD 39.64 billion in 2025 to meet global demand [1]. The principal source of palm oil is the *Elaeis guineensis* Jaq. species of palm, which is widely planted in Southeast Asia and especially in Malaysia and Indonesia. One strategy to improve palm oil production is to increase the oil extraction rate (OER) from the oil palm fruits during the milling process [2]. This can be achieved through the certification of mills and adherence to fruit grading as part of ensuring that only high-quality and ripe fresh fruit bunches are accepted and processed at all mills [3]. 

The process known as bunch analysis is traditionally used to predict the oil to bunch (O/B) ratio (%) and oil yield production (kg/palm/year) as part of the OER estimation [4]. The bunch components, mesocarp, and kernel oil content are estimated in the analysis using a traditional method devised by Blaak et al. [5] and modified by Rao et al. [6]. In this method, fruit samples are manually depericaped using a sharp knife before the mesocarp is dried in the oven and later weighed. The oven-dried mesocarp is then ground, normally using a food blender. The ground mesocarp is then sieved using a mesh before Blaak’s Soxhlet extraction takes place. Although some authors reported that the O/B figures obtained using Blaak’s method seldom agree with the OER from palm oil mills, this method is still used by most Malaysian practitioners [7]. Another problem with Blaak’s method is that it involves a Soxhlet extraction procedure, which uses n-hexane as the solvent material. N-hexane has been listed under the US toxic release inventory (TRI) since 1994 under the National Safety Council, 2007, and it is a narcotic agent that is harmful to operators [8]. Very recently, Mao et al. [9] introduced an alternative method to avoid n-hexane usage by means of a nuclear magnetic resonance (NMR) analyzer. Their studies showed promising results using a benchtop machine to enhance oil palm bunch analysis. However, their technique still required a similar fruit sample preparation to Blaak’s method, which is tedious, destructive, and time-consuming. Therefore, there is a need for a simpler technique that non-destructively predicts oil (OC) and moisture content (MC) in oil palm fruitlets.

Generally, several non-destructive quality inspection methods have been used for different types of fruit. One simple test is the floatation test, which is performed during the cleaning or sterilization of the fruits. The floatation test is a simple and efficient way of discriminating between fruits based on their specific gravity at different maturity stages. However, it is inconvenient to use on-line as it adds production processing time and costs [10]. Meanwhile, the use of other non-destructive methods, such as imaging techniques, is preferable due to their simplicity and reliable results. However, challenges still exist for practical applications, such as the time-consuming data acquisition, the high cost, the high interference from object materials and surroundings, as well as the lack of unique spectral features for different fruits and vegetables [11]. 

Electrical impedance spectroscopy has been used to monitor various agricultural products and food materials such as fruits [12,13,14], vegetables [15,16], meat [17,18], and edible oils [19,20,21]. This technique measures the electrical properties of samples as a function of a wide range of frequencies. The working principle of electrical impedance spectroscopy involves the application of an external field onto a material, such as plant tissue, to measure the material’s impedance. The results are then interpreted as the change of the electrical properties as a function of frequency [22]. In a previous work, Hamdan et al. [23] used the same technique for sludge contamination prediction in crude palm oil, and they found a prediction capability with a standard error cross-validation (SECV) of 1.04%. Similarly, Ibrahim et al. [24] used an electrical impedance spectroscopy technique for solid soluble contact (SSC) prediction in bananas, while Jamaludin et al. [25] used the same technique to indicate the plant water status in *Labisa pumila* plants.

More recently, studies have reported that electrical impedance could be used to detect resistance changes in the intracellular and extracellular compartments of fruit tissues, which may provide a method of simultaneously examining the changes that occur during ripening [26]. Due to the physical structure of the material being measured, the chemical processes within the tissue, or a combination of both parameters, the impedance may vary as the frequency of the voltage applied changes [27]. It is highly sensitive to the permeability of cell membranes and was examined on various types of fruit such as apples, bananas, Garut citruses, kiwis, lettuce, mangoes, and strawberries [28]; nevertheless, such an attempt with oil palm fruitlets has never been reported in the literature. 

Moreover, it was also stated that an electrical current is unable to cross the plant plasma membrane at low frequencies and is confined to an extracellular pathway, whereas the current will travel via the symplast at high frequencies [29]. However, high-frequency instruments are costly, while a low-frequency range is preferable in electrical system development as the instruments required for such measurements are relatively inexpensive for industrial applications [30]. Thus, this study aimed to investigate the viability of electrical impedance measurements at a range of low frequencies (<100 kHz) to predict the MC and OC of oil palm fruitlets. In comparison with the traditional Blaak method and the use of the NMR analyzer described earlier, electrical impedance spectroscopy offers advantages such as its low cost and portability, while it does not require complex pretreatment steps [31,32]. Moreover, the technique can be used for the fast, direct detection of fruit quality for on-site analysis [28,33]. Due to its tunable nature and simple design, this technique offers a simpler alternative to traditional techniques. In this research, electrical impedance spectroscopy was used to investigate the variation of the impedance spectra of oil palm fruitlets to predict the MC and OC in the fruitlets. Statistical analysis was conducted to develop prediction models to estimate the amount of MC and OC in the oil palm fruitlets. 

## 2. Results and Discussion

### 2.1. Variation of Moisture, Oil Content, and Impedance across Maturity Stages

The MC and OC results across the three maturity stages (12 WAA, 16 WAA, and 20 WAA) are shown in Figure 1. With fruit development from 12 WAA to 20 WAA, the mesocarp MC decreased linearly from around 46% to 19% (Figure 1a), while the mesocarp OC increased linearly from around 54% to 78% (Figure 1b). The changes in the mesocarp MC and OC were due to the deposition of oil in the endosperm, which began at 12 WAA and was almost complete at 16 WAA [34]. Oil deposition then continued in the mesocarp until fruit maturity at 20 WAA [35,36]. The OC increased rapidly after 12 WAA, reaching the maximum of around 78% in the fresh mesocarp at 20 WAA, which was found to be the optimal time for harvesting. The MC in the fruitlet mesocarp showed a decreasing trend during the ripening process due to the rapid accumulation of oil in the mesocarp, which is consistent with the recent findings of Suresh and Sanjib [37]. Statistical analysis of variance (ANOVA) showed significant differences between the three maturity stages for both MC and OC at *p* < 0.05 (Table 1 and Table 2).

Figure 2 shows the inverse linear regression line between the MC and OC during fruit development, with an R^2^ of 0.86. This trend is similar to that described by Ariffin et al. [38] and Hartley [39] in their studies, which explain that when oil palm fruitlets ripen, the OC inside the fruitlets reach their maximum value, while the MC reaches its minimum value. The inverse linear relationship between the MC and OC in oil palm fruitlets was also reported in the literature and can be used as a parameter to determine the mesocarp ripeness. When changes occur in the amount of moisture in the fruitlets, a change in the electrical impedance can be measured due to the sensitive and evident response of the water molecules when exposed to an alternating electrical current [40]. 

Each maturity stage produced a different percentage of MC and OC, as indicated by laboratory analysis. Table 3 shows a summary of the overall impedance measurements at four frequencies and the percentages of MC and OC at each maturity stage. The results show that the impedance value increased when the maturity stage of the oil palm fruits increased. It can also be seen from Table 3 that at 20 WAA, the highest electrical impedance values were found across all frequencies. This means that the resistivity was high and conductivity was low in the oil palm fruitlets at 20 WAA. It was observed that the MC percentage decreased from 45.92 ± 4.14% to 18.79 ± 2.36%, while the OC percentage increased from 54.02 ± 3.27% to 77.68 ± 4.38% between 12 WAA and 20 WAA due to the ripening process of the oil palm fruitlets.

### 2.2. Electrical Impedance Measurements at Different Frequencies

The impedance measurements at the four discrete frequencies of 1 kHz, 10 kHz, 20 kHz, and 100 kHz were determined for the three maturity stages of the oil palm fruitlets (12 WAA, 16 WAA, and 20 WAA) (Figure 3). All the maturity stages showed a similar electrical resistivity trend, whereby the impedance was lower at higher measurement frequencies. As shown in Figure 3, the 20 WAA maturity stage had the highest impedance, followed by those of 16 WAA and 12 WAA. These results indicate that oil palm fruitlets at later WAA stages had higher OC, lower MC, and higher impedance than oil palm fruitlets at earlier WAA stages. This was because oil palm fruitlets with high OC conduct electricity poorly as they have more resistance, thus resulting in higher impedance values. This finding agrees with the results obtained from bananas [25], mangoes [41], and apples [42], which showed declining impedance trends across the frequencies as the fruits ripened. A similar phenomenon has been observed in the electrical impedance of apple fruit tissue, which is thought to correspond to the dispersion caused by the capacitance of cell membranes [43]. Since the dielectric properties (in this case, electrical impedance) of biological tissues are considered primarily dependent on the ionic conductivity of fluids in the cellular structure and the water activity of the organisms, ionic conductivity is the factor that influences this behavior [44]. This phenomenon became the basis on which to produce statistical models from the electrical impedance measurements to predict and/or discriminate between the quality of the oil palm fruitlets. 

### 2.3. Correlation between Impedance and Moisture Content at Different Frequencies

The regression analysis results showed that all the frequencies produced similar trends regarding the linear and descendent relationships between the MC and impedance. For example, Figure 4 shows a scatter plot of the MC and impedance at 100 kHz. The regression coefficients, R^2^, between the impedance and MC were found to range between 0.34 and 0.77 (Table 4). Comparing all the frequencies, the highest regression was at 100 kHz, with an R^2^ of 0.77 (Figure 4). When the MC decreased, the ratio of water inside the oil palm fruitlets decreased, and, as a result, the impedance increased. The increase in the impedance measurement was due to the low conductivity in oil palm fruitlets. The results agree with those of previous studies [32,45]. Meanwhile, Juansah et al. [46] also reported that the impedance of Garut citrus fruits decreased with the increasing pH dilution, thus increasing the MC of the fruit. 

Based on the regression results, the correlations between the impedance and the MC of oil palm fruitlets at each frequency (1 kHz, 10 kHz, 20 kHz, and 100 kHz) were further explored to find the correlation coefficients (*r*). Statistical analysis using the Pearson correlation coefficient was performed to quantify the degree to which the frequencies correlated with the oil palm fruitlet parameters. Table 4 shows the results of the correlation between the impedance and fruitlet parameters (MC and OC) at frequencies of 1 kHz, 10 kHz, 20 kHz, and 100 kHz. Positive values indicate that both impedance and the fruitlet OC parameter were increasing. Meanwhile, negative values indicate a proportional relationship between the impedance and the fruitlet MC parameter, such that as the impedance value increased, the fruitlet parameter value decreased. The Pearson correlation coefficient values, *r,* between the MC and impedance ranged from −0.60 to −0.84. The highest negative correlation was found at 100 kHz, with an *r* of −0.84. 

### 2.4. Correlation between Impedance and Oil Content at Different Frequencies

The OC and impedance measurements were plotted to identify the correlation between both parameters. Generally, the impedance value increased as the OC increased due to the resistivity of oil as a non-polar compound. Figure 3 shows the relationship between the OC and impedance at 100 kHz. The trend with all the frequencies was broadly similar, whereby the impedance increased with higher OC. The linear regression had R^2^ values ranging from 0.33 to 0.72. From the results, the lowest R^2^ of 0.33 was found at 1 kHz (Table 5), while the highest R^2^ of 0.72 was found at 100 kHz (Figure 5). This indicates that the relationship between the OC and impedance was relatively significant at 100 kHz.

Based on the regression results, the correlations between the impedance and OC of the oil palm fruitlets at each frequency (1 kHz, 10 kHz, 20 kHz, and 100 kHz) were further explored to quantify the degree of correlation, as shown in Table 4. The Pearson correlation coefficient results for OC and impedance showed a significant correlation, ranging from 0.70 to 0.80 across the frequencies. The strongest correlation was found at 100 kHz, with a coefficient *r* of 0.80. 

### 2.5. Oil Content Prediction

The linear regression model was developed to predict the MC and OC in oil palm fruitlets for each frequency. The RMSE was also calculated based on the validation dataset tested on the linear regression models for all the frequencies (Table 5). Table 5 shows that the results of the RMSE validation values of MC ranged from 5.85% to 9.75%, while the range was 5.71% to 9.48% for the OC. According to the results, the best regression calibration equations with which to predict MC and OC were found at the frequency of 100 kHz, with an R^2^ of 0.77 and an RMSE of 5.85% and an R^2^ of 0.72 and an RMSE of 5.71%, respectively (Table 5).

Based on the linear regression model at 100 kHz, the MC and OC of the validation data set were predicted to validate the models. Figure 6 shows the R^2^ of 0.85 between the actual and prediction MC, while that of OC was found to be 0.79 (Figure 7). Therefore, this linear regression model at 100 kHz has the potential to be used to predict the MC and OC of oil palm fruitlets. 

To summarize, using a simple linear regression model to predict the MC and OC at four frequencies ranging from 1 kHz to 100 kHz and based on the experimental data showed promising impedance measurements in terms of their use for OC and MC prediction. Based on the results, the linear regression model at 100 kHz had the highest R^2^ and the lowest RMSE compared to frequencies of 1 kHz, 10 kHz, and 20 kHz for both OC and MC prediction. Thus, the linear regression model at 100 kHz was suitable for use in predicting the MC and OC in oil palm fruitlets across the maturity stages.

## 3. Materials and Methods

### 3.1. Data Collection and Sample Preparation

Oil palm fruits of the tenera variety were used in this study. They were obtained from an oil palm plantation at Universiti Putra Malaysia. The tenera species is a hybrid variety obtained by crossing the dura and pisifera genotypes, and it is often planted in Malaysia. The fruit samples were obtained from a nine-year-old oil palm (*Elaeis guineensis* Jacq.) plantation at Universiti Putra Malaysia (UPM), Serdang, Selangor, Malaysia. The plantation area has equatorial climatic conditions characterized by high temperatures, heavy rainfall, and no distinctive seasons. The annual mean temperature of the area is 38 °C, and the annual precipitation is 2000 mm. The plantation soil is classified as the Serdang series with a sandy clay texture. 

A total of 90 oil palm fruitlets were sampled at three different maturity stages, as measured by weeks after anthesis (WAA). In total, 30 fruitlet samples were acquired at 12 WAA, 16 WAA, and 20 WAA, respectively. The maturity stages of the oil palm fruits are normally indicated by color changes from black to reddish-orange, as shown in Figure 8 [45]. Only unbruised fruitlets were chosen for the experiment. The fruitlets were wiped clean and dry to minimize reading errors during the measurements. 

### 3.2. Electrical Impedance Measurements

The electrical impedance properties of the oil palm fruitlets were measured using standard disposable liquid gel silver/silver chloride (Ag/AgCl) electrocardiogram (ECG) electrodes attached to a Kelvin clip lead (16089E, Agilent, Kobe, Japan); this was connected to an LCR meter (4263B, Agilent Technologies, Kobe, Hyogo, Japan) (Figure 9). The ECG electrodes were attached to both sides of the individual fruitlets, which were fully covered by a sticky plastic foam to prevent any air gaps between the sample and the electrodes. The impedance measurement was carried out at four discrete frequencies (1 kHz, 10 kHz, 20 kHz, and 100 kHz). The LCR meter was first calibrated using the manufacturer’s standard calibration procedure to avoid any systematic errors [47]. The ECG electrodes were disposed of every three fruitlets. All the measurements were carried out at room temperature. 

### 3.3. Moisture Content Measurements

After completing the electrical impedance measurements, the MC measurements of the individual fruitlet samples were carried out. In this experiment, only the mesocarp of each fruitlet was used. The MC was measured using an oven-drying standard procedure according to the PORIM Test Method (1995). Every single fruit (mesocarp) was weighed, sliced into thin layers, and dried in a forced-air oven (UN 55 model of Memmert, Eagle, WI, USA) at 103 °C for 24 h to remove the moisture. The MC was calculated using Equation (1), expressed as a percentage of wet basis (wb).
(1)$${\%\mathrm{MC}}_{\mathrm{wb}}=\left(\frac{m_{1}-{\;m}_{2}}{m_{1}}\right)\times 100$$
where %MC_wb_ is the percentage of MC wet basis, m_1_ is the weight of the fresh mesocarp, and m_2_ is the weight of the dry mesocarp.

### 3.4. Oil Content Measurements

The oil extraction was carried out according to the PORIM Test Method (1995) using a set of Soxhlet extraction apparatus (Favorit, Puchong, Malaysia) and n-hexane (AR 99% Friendemann Schmidt, Kuala Lumpur, Malaysia) as a solvent. A rotary evaporator was used to separate the n-hexane from the extracted oil. Before the extraction, the dried oil palm mesocarp fruitlet was ground into small-sized particles using a domestic grinder (MX800S model Panasonic, Petaling Jaya, Malaysia). A known weight of the sample was packed into a filter paper (Whatman Catalogue Brand No 1001 150) and placed in a Soxhlet extraction chamber. The oil extraction process was carried out for around eight hours until the oil had been completely extracted from the samples.

Each of the six boiling flasks containing the oil and n-hexane mixture was placed in the rotary evaporator to separate the oil and n-hexane. The process took about 10 min. After the separation, the boiling flask that consisted only of the oil was heated in the forced-air oven at 60 °C for five minutes to remove the remaining n-hexane inside the oil. The boiling flask with oil was placed in a desiccator and weighed after it had been cooled to room temperature. The OC of every single fruit was calculated using Equation (2).
(2)$$\%\mathrm{OC}=\left(\frac{m_{a}-m_{b}}{m_{2}}\right)\times 100$$
where m_a_ is the weight of a flask with oil, m_b_ is the weight of an empty flask, and m_2_ is the weight of a dry mesocarp sample.

### 3.5. Statistical Analysis

The oil palm fruitlet samples were randomly divided into testing and validation sets with a 70–30% split, respectively. Analysis of variance (ANOVA) with a 5% significance level was used to analyze the impedance variation at different frequencies. The Pearson correlation was then done to determine the suitable frequency to be correlated with the MC and OC of the oil palm fruitlets. Regression analysis was used to fit and develop a linear regression model between both the MC and OC on the one hand and the impedance values on the other. The regression model was evaluated by calculating the root-mean-square error (RMSE) using Equation (3):(3)$$\mathrm{RMSE}=\sqrt{\frac{1}{N}\overset{N}{\underset{n=1}{\sum }}{\left(y_{t}-y_{e}\right)}^{2}}$$
where *N* is the number of samples in a dataset, *y_t_* is the predicted value calculated using the regression equation, and *y_e_* is the measurement value found through the experimental procedures.

## 4. Conclusions

The impedance of oil palm fruitlets was successfully measured and correlated with the MC and OC of the fruitlets. This study found a significant correlation between them, with Pearson coefficients up to −0.84 for the MC and 0.80 for the OC. In particular, at a frequency of 100 kHz, the regression coefficient, R^2^, between the impedance and both the MC and OC was 0.77 and 0.72, respectively. The RMSE of the linear regression model to predict the MC and OC was around 5.85% and 5.71%, respectively, which is promising for further sensing system development. The results indicated that electrical impedance characterization capability is instrumental in the development of a portable sensing system for a faster and simpler method of estimating the MC and OC in oil palm fruitlets, whereas the current methods used in the industry are tedious, time-consuming, and require bulky equipment or skilled personnel. This study was conducted at a generally low-frequency range (1 to 100 kHz); therefore, the instrumentation required for such measurements would be relatively inexpensive for industrial applications. The application of this method in the palm oil industry is relatively new, which confers a prospective role to this study, the general aim of which was to assess the viability of using electrical impedance spectroscopy for the in situ prediction of oil palm fruitlet quality. Such a method would be useful for monitoring oil yield production in the palm oil milling process.

## Figures and Tables

**Figure 1 plants-11-03373-f001:**
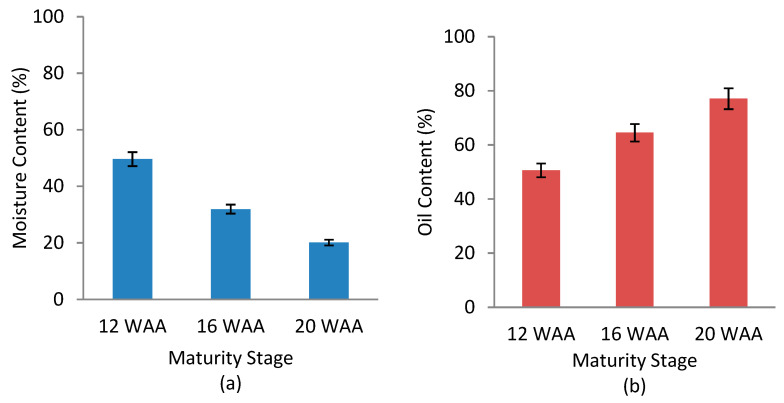
Effect maturity stages on (**a**) moisture and (**b**) oil content of mesocarp oil palm fruitlets.

**Figure 2 plants-11-03373-f002:**
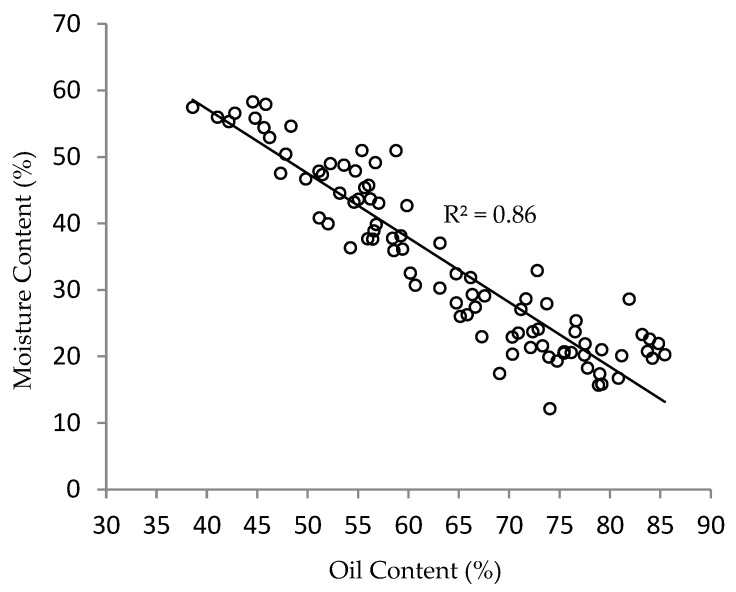
Relationship between oil and moisture content of oil palm fruitlets measured at 12, 16, and 20 WAA.

**Figure 3 plants-11-03373-f003:**
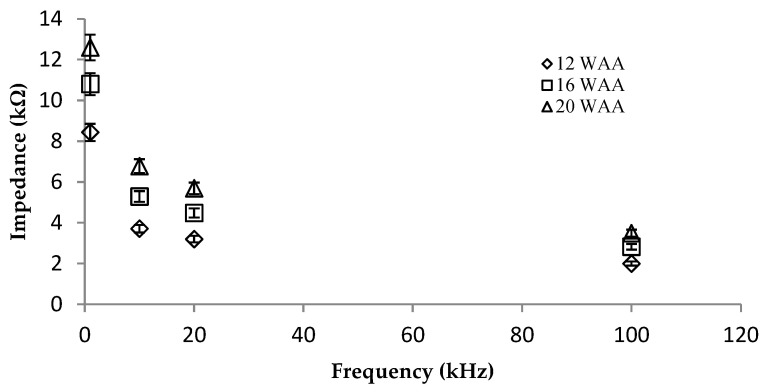
The impedance measurements of oil palm fruitlet samples at different maturity stages and different frequencies.

**Figure 4 plants-11-03373-f004:**
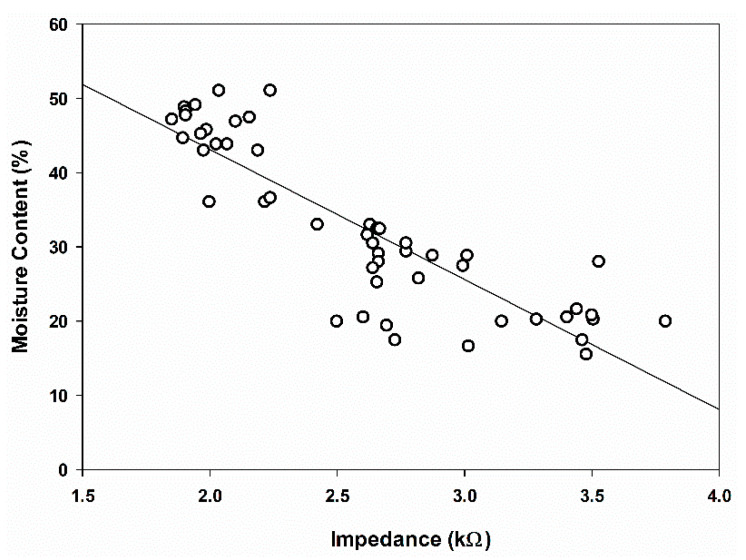
Effect of moisture content on the impedance value at 100 kHz.

**Figure 5 plants-11-03373-f005:**
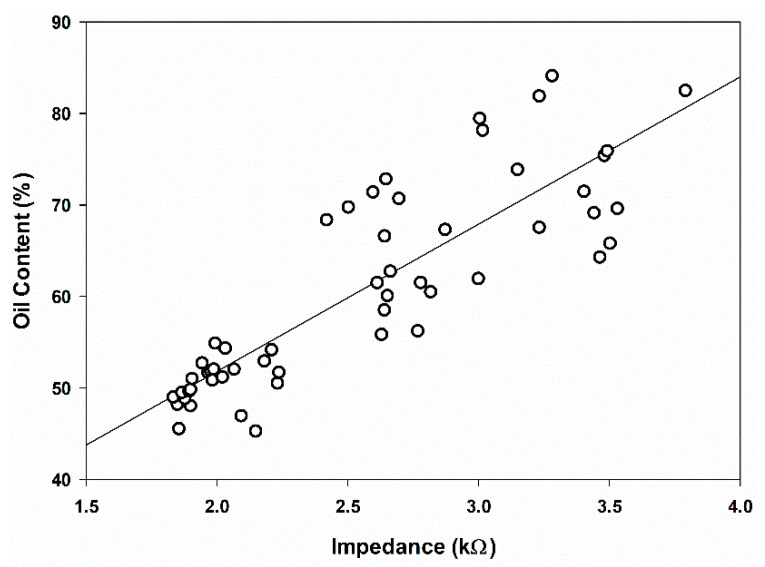
Effect of oil content on the impedance value at 100 kHz.

**Figure 6 plants-11-03373-f006:**
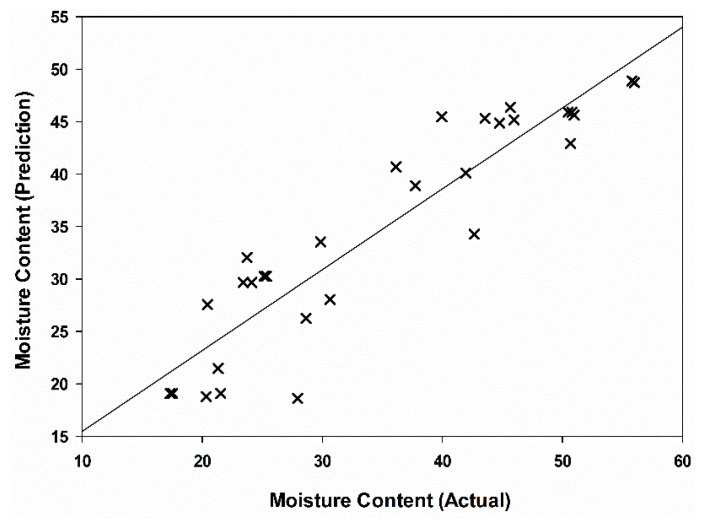
Relationship between actual and predicted moisture content based on linear calibration model at 100 kHz.

**Figure 7 plants-11-03373-f007:**
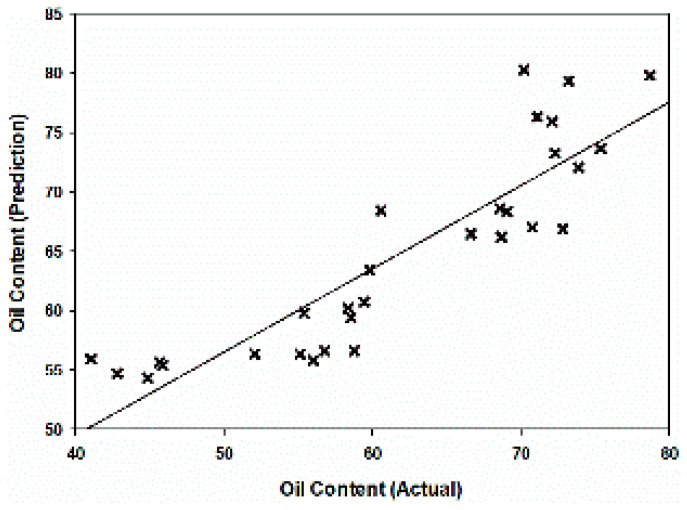
Relationship between actual and predicted oil content based on linear calibration model at 100 kHz.

**Figure 8 plants-11-03373-f008:**
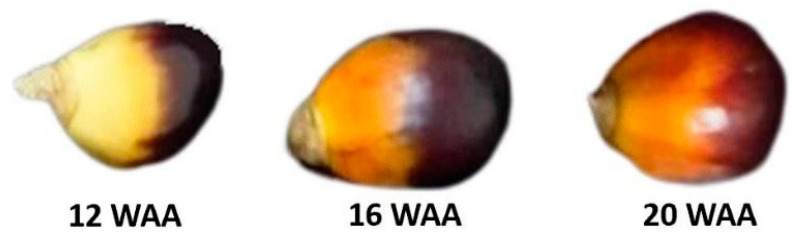
Different maturity stages of oil palm fruitlets indicated by different colors.

**Figure 9 plants-11-03373-f009:**
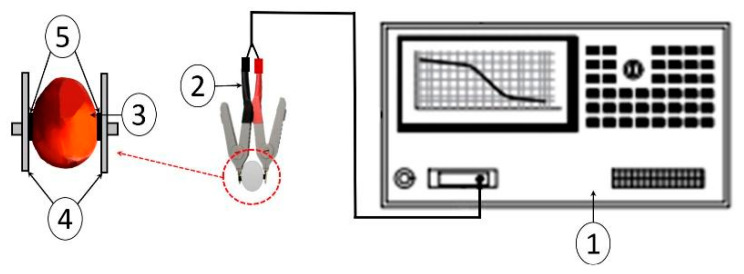
Schematic diagram of electrical impedance measurement of oil palm fruitlets; (1): LCR meter, (2): Kelvin clip leads, (3): oil palm fruitlet sample, (4): sticky plastic foam, (5): ECG electrode.

**Table 1 plants-11-03373-t001:** ANOVA results for mesocarp moisture content data.

				ANOVA	
Mesocarp Moisture Content	Sum of Squares	df	Mean Squares	F	Sig.
Between Groups *	13,053.74	2	6526.871	322.2241	0
Within Groups *	1741.989	87	20.25569		
Total	14,795.73	89			

* Groups are 12 WAA, 16 WAA, and 20 WAA.

**Table 2 plants-11-03373-t002:** ANOVA results for mesocarp oil content data.

				ANOVA	
Mesocarp Oil Content	Sum of Squares	df	Mean Squares	F	Sig.
Between Groups *	10,377.08	2	5188.541	139.604	0
Within Groups *	3196.288	87	37.16614		
Total	13,573.37	89			

* Groups are 12 WAA, 16 WAA, and 20 WAA.

**Table 3 plants-11-03373-t003:** Summary of impedance measurements at four frequency levels and the oil palm fruitlets’ properties at three different maturity stages.

Weeks after	Frequency (kHz)				Oil Content	Moisture Content
Anthesis (WAA)	1	10	20	100	(%)	(%)
12	8.43 ± 0.81	3.70 ± 0.28	3.20 ± 0.23	2.00 ± 0.12	54.02 ± 3.27	45.92 ± 4.14
16	10.80 ± 1.15	5.28 ± 0.69	4.48 ± 0.56	2.82 ± 0.33	68.14 ± 4.25	28.67 ± 3.62
20	12.59 ± 3.77	6.78 ± 1.48	5.68 ± 1.15	3.48 ± 0.66	77.68 ± 4.38	18.79 ± 2.36
Average Mean	10.61 ± 1.91	5.25 ± 0.82	4.45± 0.65	2.77 ± 0.37	66.31 ± 3.97	31.13 ± 3.37

**Table 4 plants-11-03373-t004:** Correlation coefficient (*r*) of impedance with moisture and oil content at four frequencies using the Pearson correlation method.

Parameter (%)	Frequency (kHz)			
	1 kHz	10 kHz	20 kHz	100 kHz
MC	−0.60	−0.80	−0.81	−0.84
OC	0.60	0.79	0.80	0.80

**Table 5 plants-11-03373-t005:** Linear regression model and RMSE validation for moisture and oil content.

	Frequency kHz	Linear Regression Models	Regression Coefficient R^2^	Validation RMSE (%)
Moisture Content	100	y_100_ = −17.848x + 79.083	0.77	5.85
	20	y_20_ = −10.287x + 75.614	0.73	7.43
	10	y_10_ = −7.9808x + 71.874	0.70	7.60
	1	y_1_ = −3.1296x + 64.128	0.34	9.75
Oil Content	100	y_100_ = 16.074x + 23.657	0.72	5.71
	20	y_20_ = 9.4803x + 25.88	0.71	7.25
	10	y_10_ = 7.4122x + 29.047	0.69	7.52
	1	y_1_ = 2.8869x + 36.439	0.33	9.48

## Data Availability

Not applicable.

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
