# Peer review of "Electrical Impedance Spectroscopy for Moisture and Oil Content Prediction in Oil Palm (Elaeis guineensis Jacq.) Fruitlets"

_plants, 2022, doi:10.3390/plants11233373_

Round 1

Reviewer 1 Report

1.     Key results:

This study found a significant correlation between MC and OC of the fruitlets with Pearson coefficients up to -0.84 for MC and 0.80 for OC. The impedance of oil palm fruitlets has been successfully measured and correlated with them. The results showed that electrical impedance spectroscopy has potential for MC and OC prediction in oil palm fruitlets. This evidence is useful for oil yield production monitoring in the palm oil milling process.

2.     Validity: The manuscript does not have a serious flaw, the abstract is clear, the introduction provides sufficient background, the research design appropriate, the methods adequately described, the results are clearly presented, the cited references are relevant to the research and the conclusions supported by the results. However, there is some minor mistakes.

English language and style are fine but there is minor spell check is required.

The plant name of Oil palm (Elaeis guineensis Jacq.) is not mentioned throughout the paper (Azadirachta indica) and should be written in italics in the title of the paper too.

The quality of some figures is poor and needs some enhancement and increasing resolution.

3.     Originality and significance: The conclusions are original, but needs extra support from some updated relevant references.

4.     Data & methodology: The approach is valid, and the quality of data is fine, however, the quality of presentation must be enhanced.

5.     Appropriate use of statistics: The used statistical methods are fine.

6.     Conclusions: The conclusions and data interpretation are valid, but the discussion needs extra support from some updated relevant references.

7.     References: Should be written in the same format, and there is a problem in numbering the references, no. 1 is written in no. 2, and there is a shift of references numbers throughout the manuscript and all references should be reviewed again and re-numbered in correlation with the manuscript.

Author Response

plants-1915279: Electrical Impedance Spectroscopy for Moisture and Oil Content Prediction in Oil Palm (Elaeis guineensis Jacq.) Fruitlets

Reviewer #1:

  1. Key results:

This study found a significant correlation between MC and OC of the fruitlets with Pearson coefficients up to -0.84 for MC and 0.80 for OC. The impedance of oil palm fruitlets has been successfully measured and correlated with them. The results showed that electrical impedance spectroscopy has potential for MC and OC prediction in oil palm fruitlets. This evidence is useful for oil yield production monitoring in the palm oil milling process.

Thank you for your positive feedback.

  1. Validity:The manuscript does not have a serious flaw, the abstract is clear, the introduction provides sufficient background, the research design appropriate, the methods adequately described, the results are clearly presented, the cited references are relevant to the research and the conclusions supported by the results. However, there is some minor mistakes.

English language and style are fine but there is minor spell check is required.

Agreed. The grammatical and spelling mistakes were thoroughly checked throughout the revised manuscript.

The plant name of Oil palm (Elaeis guineensis Jacq.) is not mentioned throughout the paper and should be written in italics in the title of the paper too.

Noted. The scientific name of oil palm was defined in the revised manuscript (line 34 in the Introduction section) and used in the title.

The quality of some figures is poor and needs some enhancement and increasing resolution.

We redrawn figure 1 and 2 to ensure high quality with adequate resolution. We could also submit all figures separately in high resolution to the journal editor.

  1. Originality and significance:The conclusions are original, but needs extra support from some updated relevant references.

Updated relevant references by Suresh and Sanjib (2020), You et al., (2020) and Watanabe et al., (2018) were added in the result and discussion to support the finding of this study. This is explained in detail in question 6. The conclusion section was revised accordingly to better link with other sections as also commented by other reviewers.

  1. Data & methodology:The approach is valid, and the quality of data is fine, however, the quality of presentation must be enhanced.

       By quality of presentation, we assumed this referred to the description of the methodology. We enhanced the description of the methodology by adding more details on raw materials (genotypes, pedoclimatic conditions of sampling sites, etc) which were also commented on by reviewer 4. We also redrawn Figure 1 and 2 for more clarity and carefully revised the manuscript line-by-line.

  1. Appropriate use of statistics:The used statistical methods are fine.

Thank you.

  1. Conclusions:The conclusions and data interpretation are valid, but the discussion needs extra support from some updated relevant references.

Extra support from some updated relevant references were elaborated in the results and discussion. For example, the MC in mesocarp of fruitlets showed a decreasing trend (50% to 20%) during the ripening process due to the rapid accumulation of oil in the mesocarp, which is consistent with the findings of recent work by Suresh and Sanjib [37]. The inverse linear relationship between MC and OC is similar as described by Ariffin et al., [38] and Hartley [39] in their studies which explain that when the oil palm fruitlets ripening, the OC inside the fruitlet reaches into the maximum value while the MC reaches into the minimum value. The higher impedance at higher WAA is in agreement with the results obtained from banana [23], mango [41] and apple [42] which showed declining trend of impedance across frequency as the fruits ripen. The low conductivity in oil palm fruitlets as MC decreases is in agreement with previous studies, which showed decreased impedance as MC in the material reduced [28, 39]. Also, Juansah et al., [40] reported that the impedance of Garut citrus fruits decreased with the increasing in dilution of pH thus increasing the MC of the fruit.  The conclusion section was also revised accordingly based on comments from reviewer 3 and 4 to better link with the results and discussion.

  1. References:Should be written in the same format, and there is a problem in numbering the references, no. 1 is written in no. 2, and there is a shift of references numbers throughout the manuscript and all references should be reviewed again and re-numbered in correlation with the manuscript.

Agreed. This has been corrected in the revised manuscript.

Reviewer 2 Report

A brief summary

The manuscript “Electrical Impedance Spectroscopy of Oil Palm Fruitlets” present a method based on electrical impedance measurements in order to be used as a predict the oil palm fruitlets level. The study used statistical analysis (ANOVA, mathematical linear regression and Pearson correlation) to determine the better parameters of electrical impedance spectroscopy.

It is interesting idea, but in this form the manuscript could not be suitable for publication in Plants journal, because there are major shortcomings of the method. I recommend additional investigation regarding the frequency range.

Specific comments

Comments regarding the aspect which needs to be complete revision:

Title:

1)   The title is too general. This must be fitted only on aspects presented.

Method:

2)   The investigation was performed on 1, 10, 20 and 100 kHz and the authors consider that impedance decreasing exponentially with frequency (fig.5) and the other parameters are linear. Perhaps, this supposition is true, but must be demonstrated. From 20 to 100 kHz is a large window where nonlinear behavior can appear. Thus, further investigations are needed in order to see if the linear behavior is maintained at any frequency from 1 - 100 kHz range. I recommend an additional investigate of samples, at least 3 intermediary frequencies (20-100 kHz), so that the method to be used as a general non-invasive investigation technique.

Other omissions that must be corrected:

3)   I recommend to change the notation of m1 and m2, from equation 2, in order to avoid the confusion with terms from equation 1.

4)   In sentence from lines 146-147, “…mesocarp moisture content decreased linearly from 49.62% to 20.10% (Figure 3b) while mesocarp oil content increased linearly from 50.33% to 77.09% (Figure 3a).”, the figures are reversely. In fact, the results for moisture are presented in figure 3a and for oil in figure 3b.

5)   In tables 1 and 2, the terms as Between Groups and Within Groups should be explained. I suppose that the groups mean the 12, 16, 20 WAA.

6)   In figure 4, were there included the values for all sets of data (12, 16, 20 WAA)?

 Typos:

7)   There are some typos in the text. The most common is the repetition of the same word in a sentence. For example: moisture content and oil content (line 25, 51 etc), mesocarp oil and kernel oil (line 39), oil content (Lines 146-147 etc).

8)   The word Masocarp, from table 2 (Masocarp Oil Content) should be corrected.

Author Response

plants-1915279: Electrical Impedance Spectroscopy for Moisture and Oil Content Prediction in Oil Palm (Elaeis guineensis Jacq.) Fruitlets

Reviewer #2: 

A brief summary

The manuscript “Electrical Impedance Spectroscopy of Oil Palm Fruitlets” present a method based on electrical impedance measurements in order to be used as a predict the oil palm fruitlets level. The study used statistical analysis (ANOVA, mathematical linear regression and Pearson correlation) to determine the better parameters of electrical impedance spectroscopy.

It is interesting idea, but in this form the manuscript could not be suitable for publication in Plants journal, because there are major shortcomings of the method. I recommend additional investigation regarding the frequency range.

More detailed explanations were added in the introduction section about the frequency range of this study (line 78-100) This is also to address reviewer 3’s comment about the scientific motivation of this work. It was stated that electrical current is unable to cross the plant plasma membrane at low frequency, and is confined to an extracellular pathway, whereas the current will travel via the symplast at high frequencies (Stout, 1988). However, high frequency instruments are costly, while low frequency range is preferable in electrical system development as the instrument required for such measurement is relatively inexpensive for industrial application (Khaled et al., 2015). Thus this study aimed to investigate the viability of electrical impedance measurements at a low frequency range (<100KHz) to predict MC and OC of oil palm fruitlets.

Specific comments

Comments regarding the aspect which needs to be complete revision:

Title:

1)   The title is too general. This must be fitted only on aspects presented. 

The title was revised to specify the research work, and changed to: Electrical Impedance Spectroscopy for Moisture and Oil Content Prediction in Oil Palm (Elaeis guineensis Jacq.) Fruitlets

Method:

2)   The investigation was performed on 1, 10, 20 and 100 kHz and the authors consider that impedance decreasing exponentially with frequency (fig.5) and the other parameters are linear. Perhaps, this supposition is true, but must be demonstrated. From 20 to 100 kHz is a large window where nonlinear behavior can appear. Thus, further investigations are needed in order to see if the linear behavior is maintained at any frequency from 1 - 100 kHz range. I recommend an additional investigate of samples, at least 3 intermediary frequencies (20-100 kHz), so that the method to be used as a general non-invasive investigation technique.

We agree that the measurement in between 20-100 kHz was not conducted in this study. We revised our claim on Figure 5 to confine our description in-term of discrete frequency measurements trend only. We do not see the need of conducting extra measurements on 20-100 kHz because the prediction models were developed at individual discrete frequencies. Continuous frequency measurements however are needed for prediction analysis using multispectral data which is not the main aim of this study.

Other omissions that must be corrected:

3)   I recommend to change the notation of m1 and m2, from equation 2, in order to avoid the confusion with terms from equation 1.

Agreed. This has been corrected where ma is the weight of a flask with oil, mb is the weight of an empty flask and m2 is maintained from equation 1 as the weight of a dry mesocarp sample

4)   In sentence from lines 146-147, “…mesocarp moisture content decreased linearly from 49.62% to 20.10% (Figure 3b) while mesocarp oil content increased linearly from 50.33% to 77.09% (Figure 3a).”, the figures are reversely. In fact, the results for moisture are presented in figure 3a and for oil in figure 3b.

Agreed. There are mistakes in line 146-147. This has been corrected to: mesocarp MC decreased linearly from around 46% to 19% (Figure 3a) while mesocarp OC increased linearly from around 54% to 78% (Figure 3b) and the data was checked to align with raw data in Table 3 (Line 185-188).

5)   In tables 1 and 2, the terms as Between Groups and Within Groups should be explained. I suppose that the groups mean the 12, 16, 20 WAA.

Yes. A footnote was added under the tables to explain this.

6)   In figure 4, were there included the values for all sets of data (12, 16, 20 WAA)?

Yes. The figure captions were revised to clarify this.

7)   There are some typos in the text. The most common is the repetition of the same word in a sentence. For example: moisture content and oil content (line 25, 51 etc), mesocarp oil and kernel oil (line 39), oil content (Lines 146-147 etc).

Noted. Repetition of the same word in a sentence is corrected and avoided throughout the whole manuscript.

8)   The word Masocarp, from table 2 (Masocarp Oil Content) should be corrected.

Noted. Thanks so much for the review. This has been corrected.

Reviewer 3 Report

The authors measured the electrical impedance of oil palm fruit and conducted a correlation analysis with the oil and water contents of fruit ripening. The experimental skills and data analysis are intuitive, but I cannot feel the value of any scientific innovation and potential in practical application, which is the author's stated research motivation. Otherwise, the results in this manuscript are too poor to recommend it for publication in the Plants.

Author Response

plants-1915279: Electrical Impedance Spectroscopy for Moisture and Oil Content Prediction in Oil Palm (Elaeis guineensis Jacq.) Fruitlets

Reviewer #3: 

The authors measured the electrical impedance of oil palm fruit and conducted a correlation analysis with the oil and water contents of fruit ripening. The experimental skills and data analysis are intuitive, but I cannot feel the value of any scientific innovation and potential in practical application, which is the author's stated research motivation. Otherwise, the results in this manuscript are too poor to recommend it for publication in the Plants.

It is known in literature that electrical impedance can be used to detect changes in the resistance of intracellular and extracellular compartments of fruits tissue which may provide a method of simultaneously examining changes occurring during ripening (El Khaled et al., 2017). Due to physical structure of the material under measure, or the chemical processes within the tissue, or a combination of both parameters, the impedance may vary as the frequency of the voltage applied is changing (Schröder et al., 2004). It is very sensitive to the permeability of cell membranes and was examined on various fruits such as apples, bananas, Garut citruses, kiwis, lettuce, mangoes and strawberries (Ibba et. al, 2018); nevertheless the attempt on oil palm fruitlets has never been reported in literature.

Moreover it was also stated that electrical current is unable to cross the plant plasma membrane at low frequency, and is confined to an extracellular pathway, whereas the current will travel via the symplast at high frequencies (Stout, 1988). Nevertheless, high frequency instruments are costly, while low frequency range is preferable in electrical system development as the instrument required for such measurement is relatively inexpensive for industrial application (Khaled et al., 2015). Thus this study aimed to investigate the viability of electrical impedance measurements at a low frequency range (<100KHz) to predict MC and OC of oil palm fruitlets. This explanation was added in line 87-94 in the revised manuscript.

From this study, the results indicate that electrical impedance characterization capability is instrumental in the development of a portable sensing system for faster estimation of MC and OC in oil palm fruitlets; where current traditional methods used in the industry are tedious, time consuming, requiring bulky equipment or skilled personnel.  As this study was conducted at a generally low frequency range; therefore, the instrumentation required for such measurements is relatively inexpensive for industrial application. The application of this method to the palm oil industry is relatively new, which confers a prospective role upon this study, its general aim being to assess the viability of using electrical impedance spectroscopy for in situ prediction of oil palm fruitlets quality. This paragraph was added in the conclusion section.

Reviewer 4 Report

The abstract and keywords are ok.

Too short introduction about the traditional techniques used to predict moisture content and oil content in oil palm fruitlets (add more references); the authors should focus on the main research topics and relevant questions should be addressed. For example, it is important to better highlight the advantages (and disadvantages) of your technologies in relation to the standard protocols.

In Material&Methods more information on raw materials should be added (genotypes, pedoclimatic conditions of sampling sites, etc). The analytical strategies used for this study are very innovative. Is this research only a preliminary study or do the authors think that this technology can be considered at the top? Please add this information to the title.

In the result section, the authors should better compare their results with other techniques used for similar purposes. Integrating the Results and Discussion sections could be important to avoid repetitions and better highlight the results of this study.

The conclusion is clear in relation to the study, but it should be linked in a better way to the other parts of the paper. Delete redundant information, please.

Author Response

plants-1915279: Electrical Impedance Spectroscopy for Moisture and Oil Content Prediction in Oil Palm (Elaeis guineensis Jacq.) Fruitlets

Reviewer #4: 

The abstract and keywords are ok.

Too short introduction about the traditional techniques used to predict moisture content and oil content in oil palm fruitlets (add more references); the authors should focus on the main research topics and relevant questions should be addressed. For example, it is important to better highlight the advantages (and disadvantages) of your technologies in relation to the standard protocols.

More explanation of the traditional techniques used to predict moisture and oil content in oil palm fruitlets were added. In this method, fruits samples are manually depericaped using a sharp knife before the mesocarp is dried in the oven and later weighed. The oven-dried mesocarp is then ground, normally with a food blender. Then the ground mesocarp is sieved using a mesh before the Blaak’s Soxhlet extraction is taking place. In comparison with the traditional method of Blaak’s and NMR analyzer described in the manuscript, electrical impedance spectroscopy offers advantages such as low cost, portability, and does not require complex pretreatment steps (Chowdhury et al., 2018; Grossi and Riccò, 2017). Moreover, the technique can be used for fast and direct detection of fruit quality for on-site analysis (Ibba et al., 2018). Due to their tunable nature and simple design, this method offers a simpler alternative to the traditional techniques. This elaboration was added in the introduction section (line 95- 100).

In Material&Methods more information on raw materials should be added (genotypes, pedoclimatic conditions of sampling sites, etc).

Oil palm fruits of the tenera variety were used in this study. The tenera species is a hybrid variety obtained from the cross between dura and pisifera genotypes which is often planted in Malaysia. The fruit samples were obtained from a 9-year-old oil palm (Elaeis guineensis Jacq.) plantation at the Universiti Putra Malaysia (UPM)  Serdang, Selangor, Malaysia. The plantation area has equatorial climatic conditions, characterized by high temperature and heavy rainfall with no distinctive seasons. The annual mean temperature for the area is 38°C and the annual precipitation is 2000 mm. The plantation soil is classified as Serdang series with a sandy clay texture.

The information above was added in the Material and Methods section.

The analytical strategies used for this study are very innovative. Is this research only a preliminary study or do the authors think that this technology can be considered at the top? Please add this information to the title.

Agree. We had revised the title accordingly.

In the result section, the authors should better compare their results with other techniques used for similar purposes. Integrating the Results and Discussion sections could be important to avoid repetitions and better highlight the results of this study.

Our results on the correlation between the MC and OC were compared with Ariffin et al., [38] and Hartley [39] who also found the same correlation by using traditional lab analysis. We agreed that integrating the Results and Discussion sections could avoid repetitions and better highlight the result, thus in the revised manuscript, these sections were integrated (page 5-7) and become Section 3. Results and Discussion.

The conclusion is clear in relation to the study, but it should be linked in a better way to the other parts of the paper. Delete redundant information, please.

The conclusion was revised and carefully checked to avoid redundant information. Additional statements (line 356-369) were added to better link the scientific innovation and practical application.

Round 2

Reviewer 2 Report

Dear authors

I saw your comments and I have nothing to add. I consider that the manuscript is suitable for publication in Plants journal in this form.

Author Response

Dear authors

I saw your comments and I have nothing to add. I consider that the manuscript is suitable for publication in Plants journal in this form.

Thank you very much for your constructive comments.

Reviewer 3 Report

Thanks to the authors for their responses and revisions to the manuscript, especially in the narrowing of the title topic and research motivation. My focus is on the feasibility of this technology application. The oil content ratio and maturity of objects can be screened by "Buoyancy Test" in the stage of cleaning the appearance of agricultural products. The article suggests that the simple physical evaluation method commonly used in the industry can be discussed. (flotation and appearance inspection, image analysis) its advantages and disadvantages, and then explain the application advantages of the electrical impedance method.

Author Response

Thanks to the authors for their responses and revisions to the manuscript, especially in the narrowing of the title topic and research motivation. My focus is on the feasibility of this technology application. The oil content ratio and maturity of objects can be screened by "Buoyancy Test" in the stage of cleaning the appearance of agricultural products. The article suggests that the simple physical evaluation method commonly used in the industry can be discussed. (flotation and appearance inspection, image analysis) its advantages and disadvantages, and then explain the application advantages of the electrical impedance method. 

Thank you very much for your constructive suggestion. We've added the discussion on using the floatation test and imaging methods in lines (62-72) and relevant references in the revised manuscript.
